# Genetic Parameter and Hyper-Parameter Estimation Underlie Nitrogen Use Efficiency in Bread Wheat

**DOI:** 10.3390/ijms241814275

**Published:** 2023-09-19

**Authors:** Mohammad Bahman Sadeqi, Agim Ballvora, Said Dadshani, Jens Léon

**Affiliations:** 1INRES-Plant Breeding, Rheinische Friedrich-Wilhelms-Universität Bonn, 53113 Bonn, Germany; mbsadeghi1@gmail.com (M.B.S.); j.leon@uni-bonn.de (J.L.); 2INRES-Plant Nutrition, Rheinische Friedrich-Wilhelms-Universität Bonn, 53113 Bonn, Germany; dadshani@uni-bonn.de

**Keywords:** genetic parameter, hyper-parameter, genomic selection model, estimation, nitrogen use efficiency and wheat

## Abstract

Estimation and prediction play a key role in breeding programs. Currently, phenotyping of complex traits such as nitrogen use efficiency (NUE) in wheat is still expensive, requires high-throughput technologies and is very time consuming compared to genotyping. Therefore, researchers are trying to predict phenotypes based on marker information. Genetic parameters such as population structure, genomic relationship matrix, marker density and sample size are major factors that increase the performance and accuracy of a model. However, they play an important role in adjusting the statistically significant false discovery rate (FDR) threshold in estimation. In parallel, there are many genetic hyper-parameters that are hidden and not represented in the given genomic selection (GS) model but have significant effects on the results, such as panel size, number of markers, minor allele frequency, number of call rates for each marker, number of cross validations and batch size in the training set of the genomic file. The main challenge is to ensure the reliability and accuracy of predicted breeding values (BVs) as results. Our study has confirmed the results of bias–variance tradeoff and adaptive prediction error for the ensemble-learning-based model STACK, which has the highest performance when estimating genetic parameters and hyper-parameters in a given GS model compared to other models.

## 1. Introduction

Estimation and prediction play a key role in breeding programs. For a long time, breeders have tried to predict better genetic performance, genotypic value (GV) or breeding value (BV) from observations of a phenotype of interest by using estimators of genetic and phenotypic variance. This ratio is usually the heritability based on the line mean [1]. Single nucleotide polymorphisms (SNP) with a microarray platform has become the most popular high-throughput genotyping system in recent decades, and has been extensively used for quantitative trait loci (QTL) and experimental population analysis [2,3,4,5]. Thousands of QTLs inheriting simple traits of agronomic importance have been identified in major crops, and these can be used to accelerate marker-assisted selection (MAS). However, the genetic improvement of complex quantitative traits by using QTL-associated markers or MAS is not very efficient in practical breeding programs due to QTL × environment interactions or variation in genetic population structure. MAS is effective only for alleles with large effects on quality traits. However, it cannot improve polygenic traits, and many important traits in plant breeding are polygenic [6,7]. However, genomic selection (GS) uses high-density markers to predict the genetic values of genotypes, which is different from QTL analysis, MAS and association mapping (AM). With the availability of cheap and abundant molecular markers, genomic selection (GS) is becoming an efficient method for selection in both animal and plant breeding programs. Currently, phenotyping is still expensive, requires high-throughput technologies and is a very time-consuming process compared to genotyping. Therefore, researchers are trying to predict phenotypes based on marker information. GS consists of genomic file (SNPs) and phenotype file (individuals) in the reference (training) population and predicts the phenotypes or breeding values (BVs) of candidates for selection in the test (validation) population using statistical machine learning models [8]. The training set combines genomic data as independent variables with the agronomic trait of interest as dependent variables. The density of markers is very high and contains enough information to train the GS model with greater accuracy. However, the test set contains only the genomic data of some individuals and predicts the BVs of individuals according to the GS model. A higher correlation between the predicted BVs and the true phenotypic values of the individuals implies higher accuracy and performance. Basically, there are two type of features in the GS model including genetic parameters with random effects and hyper-parameters with fixed effects, which determine the results. Genetic parameters such as population structure, genomic relationship matrix, marker density and sample size are the major factors that increase the power and accuracy of the model. However, they play an important role in adjusting the statistically significant false discovery rate (FDR) threshold in estimation. For example, mixed populations [9,10], copy numbers of variants with small effect sizes [11], rare alleles in linkage disequilibrium (LD) decay derived from marker information [12] and outliers in phenotypic observations of interesting complex traits can lead to casual signals and pseudo association between marker and trait. In parallel, there are many genetic hyper-parameters that are hidden and are not represented on a given GS model, but have significant effects on the results, such as panel size, number of markers, minor allele frequency (MAF), number of call rates for each marker, number of cross validations (CV), and batch size in the training set of the genomic file [13,14]. However, GS models face a range of practical and theoretical problems in estimating the genetic parameters and hyper-parameters of the model. The main challenge is being sure of the reliability and accuracy of the predicted BVs as results. GS via linear mixed regression is based on conventional point estimators such as maximum likelihood (ML) and restricted ML (REML), which are generally susceptible to estimating genetic parameters in the whole genomic dataset because of high collinearity in the model. Therefore, the model introduces a strong estimation bias. Consequently, QTLs with small effects are completely missed in the results [2,15]. Moreover, high variance due to high convergence under marker density is a problem that often occurs when implementing complex mixed linear GS models [16]. Recent developments in shrinkage estimation [17] and the utilization of Markov Chain Monte Carlo (MCMC) sampling methods have made GS based on Bayesian whole genome regression feasible. Nevertheless, MCMC sampling algorithms can suffer from slow convergence rates and poor mixing of sample chains [18], especially when non-additive genetic random effects are included in the model [19]. In GS, the use of high-density markers requires the application of advanced feature selection algorithms. In Bayesian whole genome regression, even shrinkage algorithms cannot provide an acceptable tradeoff between bias and variance due to the high convergence rate under high marker density [20]. Using modern statistical models could be a logical solution to this challenge. In recent decades, new technologies such as sensors, robotics and satellite data have led to a high throughput of phenotypic data. In parallel, next generation sequencing (NGS) techniques have made it possible to simultaneously generate a large training dataset for traits of interest. Thus, GS with large genomic datasets and high-density markers as features in the model, requires statistical machine learning methods with more computational power, especially for complex traits such as nitrogen use efficiency (NUE) in wheat [14,21].

### 1.1. GS Model Definition

In the rrBLUP model, *GEBV* = $X\hat{g}$, which can be considered as a regularization parameter. So,
$$\left[\begin{array}{c} \hat{\mu }\\ \hat{g} \end{array}\right]={\left[\begin{array}{cc} 1_{n}*1_{n} & 1_{n}*X\\ X^{*}1_{n} & X^{*}X+I\frac{\sigma_{e}^{2}}{\sigma_{g}^{2}} \end{array}\right]}^{-1}\left[\begin{array}{c} 1_{n}*y\\ X^{*}y \end{array}\right]$$

The $I\frac{\sigma_{e}^{2}}{\sigma_{g}^{2}}$ component in the rrBLUP matrix occurs in theory, but in practice it is required to be adjusted to a more accurate parameter such as the $G^{-1}\frac{\sigma_{e}^{2}}{\sigma_{v}^{2}}$ component in the gBLUP matrix:$$\left[\begin{array}{c} \hat{\mu }\\ \hat{v} \end{array}\right]={\left[\begin{array}{cc} 1_{n}*1_{n} & 1_{n}*Z\\ Z^{*}1_{n} & Z^{*}Z+G^{-1}\frac{\sigma_{e}^{2}}{\sigma_{v}^{2}} \end{array}\right]}^{-1}\left[\begin{array}{c} 1_{n}*y\\ Z^{*}y \end{array}\right]$$

In the LASSO model, the regularization parameter is constrained by minimum ordinary least squares (OLS) of the large genomic dataset. The GS models rrBLUP and gBLUP face incredible biases in *GEBV* calculation due to collinearity. To obtain a solution for LASSO, the $X^{*}X+\lambda I$ component is updated so that either marker effect with addition or subtraction can be computed per each iteration. The SNP effects were calculated as
$$\left[\begin{array}{c} \hat{\mu }\\ \hat{\beta } \end{array}\right]={\left[\begin{array}{cc} 1_{n}*1_{n} & 1_{n}*X\\ X^{*}1_{n} & X^{*}X+\lambda I \end{array}\right]}^{-1}\left[\begin{array}{c} 1_{n}*y\\ X^{*}y \end{array}\right]$$

*GEBV*s for each individual BGLR matrix are estimated as follows:$$\left[\begin{array}{c} \hat{\mu }\\ \hat{\begin{array}{c} g_{1}\\ .\\ .\\ .\\ \hat{g_{p}} \end{array}} \end{array}\right]=\left[u\right]{\left[\begin{array}{ccc} 1_{n}*1_{n} & \;\;\;\cdots & 1_{n}*X_{1}\\ \vdots & \;\;\;\;\ddots & \vdots \\ {X_{p}}^{*}1_{n} & \;\;\;\;\;\cdots & {X_{P}}^{*}X_{P}+I\frac{\sigma_{e}^{2}}{\sigma_{gp}^{2}} \end{array}\right]}^{-1}\left[\begin{array}{c} 1_{n}*y\\ .\\ .\\ .\\ {X_{P}}^{*}y \end{array}\right]$$

In the BGLR model, the coefficients of marker effects were tested under the following hypotheses:$$H_{0}:\;g_{1}=g_{2}=g_{n}=0$$
$$H_{a}:\;g_{1}\neq g_{2}\neq g_{n}\neq 0$$

Thus, the u value, as Bayes factor and regularization parameter, is an index for testing these hypotheses. When the number of features (p) is much higher than the number of observations (n), the challenge is to find optimal hyper-parameters for the given GS model that play a dominant role in GS. However, in these four methods, SNP selection is performed via a nonlinear transformation, typically achieved using the kernel trick. The kernel (RKHS and SVM) or ensemble (boosting and bagging) approaches allow feature selection in the n-dimensional space of SNPs with a random number of predictors p [22].

In the RKHS model, if $g(x_{i})$ is a nonparametric function of large marker density, it can be written as
$$f\left(g\left(x_{i}\right)|\lambda \right)=\frac{1}{2}\left[y-W\theta -g\left(x_{i}\right)\right]R^{*-1}+{\frac{\lambda }{2}|g\left(x_{i}\right)|}_{H^{*}}^{2}]$$
where y is the *NUE* vector, W is the incidence matrix of parametric SNP effects θ on y, R is the residual covariance matrix, $g\left(x_{i}\right)$ is the vector of genotypes (SNPs), λ is the regularization parameter under squared norm of $H^{*}$ as Hilbert space. Thus, λ controls the tradeoff between goodness and complexity of the model [23].

The SVM model can be interpreted as a class of kernel algorithms, which is very similar to the RKHS model. It is written as
$$f\left(g\left(x_{i}\right)|\lambda \right)=\frac{1}{2}V\left[y,g\left(x_{i}\right)\right]R^{*-1}+{\frac{\lambda }{2}|g\left(x_{i}\right)|}_{H^{*}}^{2}]$$
where $V\left[y,g\left(x_{i}\right)\right]$ is the insensitive loss function of support vectors for y (*NUE* vector) under covariance matrix (R), and λ is the regularization parameter under the squared norm of $H^{*}$ as Hilbert space.

In the Boosting model, the *GEBV* for *NUE* among each genotype is represented as follows:$$GEBV=\sum_{L=1}^{L_{i}}{vh}_{z}\left(g_{ens}\left(x_{p}\right)|W\right)+Bias\left(g_{ens}\left(x_{p}\right)\right)+e$$
where $L_{i}$ is the learning rate of predictors, v is the regularization parameter, $h_{z}$ is the accuracy mean of the GS model, W is the incidence matrix of SNP effects on the ensemble estimator of $g_{ens}\left(x_{p}\right)$ with the expected function and e is the residuals with independent and identity distribution (IID).

In the Bagging model, the BLUE estimator is represented as follows:$$GEBV=\frac{1}{L}{\hat{\;g}}_{Bag}\left(ij\right)E\left[{\hat{g}}_{ens}\left(i.\right)\right]+Var({\hat{g}}_{ens}\left(.j\right))$$
where $L_{i}$ is the learning rate of predictors, ${\hat{g}}_{Bag}\left(ij\right)$ is the bagged estimator based on allele frequency of the genotype, ${\hat{g}}_{ens}\left(..\right)$ is the ensemble estimator of the model that can be considered as a regularization parameter, $E\left[{\hat{g}}_{ens}\left(i.\right)\right]$ is the bootstrap mean as an estimation of bias of ${\hat{g}}_{ens}\left(..\right)$.

The stacking model combines all GS models through meta-learning. It is presented as follows:$${Loss}_{GEBV}=\left\{\begin{array}{c} \;\;\;0\;\;\;\;\;\;\;\;\;\;\;\;\;\;\;\;\;\;\;\;\;\;\;\;\;\;\;\;\;\;\;\;\;\;\;\;\;\;\;\;\;\;\;\;if\;f\left(x\right)<\varepsilon \;\\ \left|f\left(x\right)\right|-\varepsilon \;\;\;\;\;\;\;\;\;\;\;\;\;\;\;\;\;\;\;\;\;\;\;\;\;\;\;\;\;\;if\;f\left(x\right)=\varepsilon \\ \left|f\left(x\right)+\left(Bias\pm Var\right)\right|\;\;\;\;\;\;\;\;\;\;if\;f\left(x\right)>\varepsilon \end{array}\right.$$

So, $f\left(x\right)=\sum_{L=1}^{L_{i}}(m_{.j}\alpha_{.j})$ and $\left(x\right)\sim \hat{y}$.

$f\left(x\right)\sim \hat{y}$ is the predicted *NUE* value among each genotype, ε is the residual error of *GEBV* estimation, $L_{i}$ is the learning rate of predictors in a given GS model, $\left(Bias\pm Var\right)$ is the tradeoff between bias and variance of the *GEBV* estimation, ${allel}_{.j}$ is the .jth MAF for SNP and $\alpha_{.j}$ is .jth SNP effect based on the given GS model.

### 1.2. Feature Selection in GS Model

Modern statistical genomics algorithms utilize high-dimensional genomic data to perform customized and accurate genomic prediction and selection. In these algorithms, feature selection is the key step for analyzing high-dimensional genomic and phenotypic data simultaneously. Recent advances include next generation sequencing (NGS) and high-throughput phenotyping techniques that generate a large number of variables in different types of GS models. This development of complex data structures leads to structured identification of important features in the model. This big dataset can be considered as a matrix, with columns corresponding to variables such as SNPs and explanatory phenotypic traits, and rows corresponding to individuals. Since the number of measured features is much larger than the number of individuals, this high dimensionality of the dataset has led to heterogeneous feature selection [24,25]. The question arises of how to identify the important features among the trait of interest from this large-scale data [26]. Breeders often measure multiple variables in each genotype simultaneously. Therefore, multivariate data are very common due to the facility of data collection. Therefore, for complex features, the relationship between individuals is nonlinear. Defining a complex model is usually the solution against poor accuracy, especially for the GS model, which is inherently nonlinear, with parametric and some semi-parametric estimators. Feature selection summarizes the variables into a small subset. Two complementary items include predictive performance and stability of the selected features, which makes up an acceptable feature selection method. Simple feature selection algorithms rely only on univariate GS models such as single locus prediction. However, in practice, most genomic datasets are multivariate with different classes and probability distributions [27]. Thus, one challenge is to identify the subset of variables that are useful in a given prediction GS model. The interpretability of selected variables is important for the stability of the given model, which is related to the heterogeneous nature of genomic data [28,29]. Indeed, marker information with minor or major alleles, heterozygous and missing data are highlighted at the categorical scale, while agronomic (phenotypic) traits may be on continuous or categorical scales. Currently, there are few feature selection methods that directly handle both continuous and categorical variables. In general, kernel methods and tree ensemble approaches are common for semiparametric GS models with both continuous and categorical variables [30]. A mixed linear GS model with high marker density leads to nonlinear regression in the surrounding space. This is typically achieved via the kernel method, which allows the computations to be performed in the n dimensional space of variables for any number of predictors p. The kernel method is actually kernel smoothing of the mean, which is a set of tools used to perform non- or semi-parametric estimation [31]. Multi-trait reproducing kernel Hilbert space (RKHS) methods can be modeled with marker–environment interaction; therefore, they are suitable for continuous response variables such as NUE with unknown prior distribution [32]. Tree ensemble approaches such as random forest (RF) provide a better generalization performance because, in these approaches, the errors of the estimators (e.g., BLUE) are distributed across different decision trees [1,33]. In general, the tree strategy attempts to minimize covariate error when approximating the true class distribution while shrinking the effects of known factors to null [34].

### 1.3. Regularization of GS Model

The main objective of genomic prediction (GP) approaches is to estimate genotypic values among unobserved true phenotypic values. However, determining the relevant predictive genomic estimated breeding values (*GEBV*s) based on high-density marker information is a fundamental problem in GS models, especially for complex quantitative trait with large number of SNPs (*n*) and very small *p-*values [35,36]. SNP heritability estimation takes the role of regularization, but how can one tell whether these estimated BVs are good or bad? Before providing an answer to this question, it should be clarified that GP is different from genomic inference. GP can be empirically calibrated, but inference cannot. Therefore, the regularization approach is an important step towards correct inference [37]. Deep learning (DL) algorithms attempt to estimate any minor or major genetic effects. The utilization of DL algorithms in GS provides the opportunity to obtain a meta-picture of the whole GS performance [38]. Even when using DL algorithms as a modern statistical perspective, GS models suffer from under- or over-fitted results. Therefore, regularization techniques provide a balance between under- or over-fitting in the GS model. That is, regularization provides a set of tools to find a logical tradeoff between bias and variance of the parameters and semi-parameters of the estimated genetic gain in the GS model. Other GS models with kernel, Bayesian or DL roots have been derived from Equation (1). Genetic gain in specific or estimated breeding value (EBV) in general can be defined by the following formula [39]:(1)$$∆G=g_{si}\beta_{i}\sigma_{e}$$
where ∆G is the genetic gain based on marker information, $g_{si}$ is the genetic selection density in the population, $\beta_{i}$ is the power of the equation based on the FDR threshold of significant markers and $\sigma_{e}$ is the genetic standard error (SE) or the residuals of the equation. Once high-density markers are used in the GS model, computing $g_{si}$ with many pairwise hypothesis tests at the same time has a high bias and high variance. Therefore, modern feature selection such as kernel or ensemble tree approaches could be a solution to this challenge. The $\beta_{i}$ can be defined as the matrix below:(2)$$\sum \beta_{i}=\log\;\left[\begin{array}{cc} X^{\prime }X & X^{\prime }Z\\ Z^{\prime }X & 1+\frac{1^{G^{*-1}}}{\sigma_{e}^{2}} \end{array}\right]$$ where X is the incidence matrix for the proportion of individuals in the population structure (n_ps_) × marker (m) with fixed effects, $X^{\prime }$ is a transformation of X, Z is a designed matrix for the effect of genotype (n) × marker effect (p), including all random effects, $Z^{\prime }$ is the transformed Z, $G^{*-1}$ is the inverted matrix of the genomic relationship matrix (GRM) when the effect of non-associated markers has shrunk toward zero with *N*(0,$\;\sigma_{e}^{2}$) and $\sigma_{e}^{2}$ is the covariate error of the GS model in the form of BLUEs. $\beta_{i}$ is the power of the GS model, and it can be considered as a regularization parameter when fitting a neural network (NN) GS model. The regularization parameter is a function of tradeoff between bias and variance. This clearly indicates that the GS model is over-fitting or under-fitting the training set. Other parameters of the model are controlled by the regularization parameter. In the network, an input layer with large weights can lead to large changes in the output layer as a result of the model [39]. In this situation, parameter estimation and model performance ($\beta_{i}$) are likely to be poor for new data. Therefore, when using modern statistical algorithms such as kernel or tree ensemble, the weights in the input layers are kept small [40]. Feature selection and regularization are two useful concepts in situations where, in the GS model, the number of genetic parameters is much larger than the number of observations. Feature selection is required to prevent over-fitting or under-fitting of selection or classification models, and it minimizes the computation time and loss function error of the model [1,41]. Regularization attempts to account for genetic hyper-parameters such as number of clusters in the population structure, the effect of MAF on genomic relationship matrix (GRM), LD decay and SNP covariate effects [42]. Thus, this approach leads to a higher performance in estimation, when the regularization parameter is very well defined in the given GS model. In this study, we evaluate different GS models and their predictive ability to improve prediction accuracy in the context of a phenotypic and genotypic NUE dataset of 221 bread wheat genotypes among three classes of regression learning methods, kernel and ensemble algorithms. We emphasize that the focus of this study is to compare GS methods based on ensemble learning algorithms and regression approaches with the aim of (i) optimizing genetics parameters and hyper-parameters of the population in the given GS model, (ii) identifying an appropriate regularization parameter for a given specific GS problem and (iii) demonstrating the performance of the best GS model through bias–variance analysis and error measurement.

## 2. Results

### 2.1. Genetic Parameters and Hyper-Parameter Estimation

Genetic parameter estimates and their 95% CL bootstraps for the *NUE* vector of 221 bread wheat genotypes based on all GS models separately. SNP heritability estimates were derived from random SNP effects in the given GS model. The highest and lowest SNP heritability estimates were related to the STACK model (0.62) and the gBLUP model (0.28), respectively, at low N levels. At the HN level, the highest SNP heritability was found with the STACK model (0.71) and the lowest with the gBLUP (0.30) and BGLR (0.30) models. In both the training phase with CV_K-fold_ = 10 and the testing phase with CV_K-fold_ = 5, the STACK model, which is based on ensemble learning inference, had the highest *GEBV* mean under low and high N levels, at 0.69 and 0.76, respectively. For the rrBLUP, gBLUP, BGLR, RKHS and SVM models, differences in hypothesis testing between the *GEBV* means in the training and testing phases were significant at low and high N levels (*p*-value with α=0.05), indicating that the performance of these models is worse than against other inferences, such as ensemble learning GS models. Genetic hyper-parameter estimates including learning rate, number of iterations and batch size for the *NUE* vector of 221 bread wheat genotypes based on all GS models separately are shown in Table 1. Based on the regularization parameters of each model, the minimum learning rate of 0.01 was computed for the rrBLUP, LASSO, RKHS and BOOST models, using the rule $\alpha_{j}=\frac{100\alpha_{0}}{100+j}$, where $\alpha_{0}$ is the initial learning rate 1 and j is a counter of epochs up to 9900. For the remaining GS models, the minimum learning rate of 0.001 was calculated. The comparison between accuracy (%) in both the training phase with CV_K-fold_ = 10 and the testing phase with CV_K-fold_ = 5, among all ensemble learning algorithms, including BOOST, BAGG and STACK, was not significant, indicating that the accuracy of these models is higher than other GS models with kernel and linear algorithms.

### 2.2. Bias–Variance Tradeoff in GS Models

The bias–variance tradeoff analysis for the *NUE* vector generated from the 150 K affymetrix SNP Chip, under low and high N is shown in (Table 2 and Table 3). As can be seen, the loss value of a given GS model using the Scikit-learn algorithm indicates an irreducible error that is constant at both low and high N levels, and it is possible to minimize and control the effect of hyper-parameters in the model that were not defined in the given model. The effects of the main genetic parameters depending on the definition of the given model are clearly shown in Figure 1a,b. Based on the genetic structure and kinship of the population after model regularization, K = 3 was determined as the optimal number for model complexity analysis. At low N levels, both SVM and RKHS models with kernel inference exhibited the highest and lowest bias and variance, respectively. Thus, these models may represent upper and lower thresholds for the bias–variance tradeoff. At high N levels, the SVM model with kernel inference and rrBLUP with frequentist linear inference had the highest and lowest bias and variance, respectively. It can be concluded that the interaction between N level and wheat genotypes based on SNP information is significant. At both low and high N levels, the behavior of the BOOST and BAGG models with ensemble learning inference showed a moderate tradeoff between under and upper fit. Thus, after mean comparison (LSD (0.05)), we can conclude that the bias, variance and average of expected loss between the GS models overall resulted in statistically significantly different means. This difference indicates that the performance of some models to predict *GEBV*s is higher than others.

### 2.3. Error Measurement of GS Models

Measurement error in *GEBV* predictors causes bias in estimated genetic parameters in GS models. Thus, error measurement of GS models clarifies the origin of this error, which is related to model definition or variables in the model, such as marker information and phenotypic values. After performing the bias–variance analysis to obtain an overall view of the performance of GS models, the adaptive standard error of prediction for each model under low and high N levels, were pairwise compared (Figure 2). Under low N levels, the BAGG model, and at high N levels, the STACK model, showed the lowest error in predicting the genomic parameters. The lowest error in prediction confirms the result of the bias–variance analysis on these two models with ensemble learning inference. Probably due to collinearity in the whole genomic regression analysis, rrBLUP has the highest error in the prediction at both low and high N levels compared to other models (Appendix A). Therefore, STACK can be selected as the best GS model with a high performance for predicting *GEBV*s.

### 2.4. Genetic Selection Gain Estimation Based on Selected Model

Genetic selection gain was estimated using the genetic value of genotypes in the GRM and STACK models. For this purpose, the wheat population was divided into training set (75%) and testing set (25%) genotypes. As can be observed in Figure 3a, the *NUE* values (%) predicted based on *GEBV*s are positively correlated with the actual *NUE* values, as the regression slope in the training set is positive at both low and high N levels. Additionally, the distribution of genotypes around the regression line is almost the same in the training and testing sets, indicating that the accuracy of the STACK is a good fit to the predicted values of *NUE* (%) at both low and high N levels. In Figure 3b, the RE (%) of the top ten genotypes with the highest *GEBV*s based on the STACK model is shown. Six genotypes with high RE (%) were replicated at both low and high N levels. Therefore, allelic variation in these top six genotypes was plotted against the minor and major alleles of the entire population (Figure 4). As can be observed, there is a significant difference in allelic content between the average of the top six genotypes and the minor alleles in the whole population, but there is no difference in the major alleles, and they are in the same group. Since the regularization parameter in this model is MAF with bias and variance, it could find more than 50% of the top genotypes with high accuracy.

## 3. Discussion

Currently, phenotyping is still expensive, requires high-throughput technologies, and is a very time-consuming process compared to genotyping. The use of modern statistical models could be a logical solution to this challenge. In recent decades, the use of new technologies such as sensors, robotics and satellite data has led to high-throughput phenotypic data. In parallel, new techniques such as next generation sequencing (NGS) have made it possible to create a big training dataset for the trait of interest. Thus, GS with a big genomic dataset and high-density markers as features in the model requires statistical machine learning methods with more computational power, especially for complex traits such as nitrogen use efficiency (NUE) in wheat [14,21]. Many machine learning methods have been adapted and developed for GP and GS [1,43]. In particular, several important parametric models, such as neural networks, kernel regression and ensemble learning algorithms have been generalized to handle high-density SNP data [38,44]. They provide GS studies with more comprehensive and flexible methods to estimate *GEBV*s with high accuracy. For the *GEBV*s to achieve high accuracy, all the assumptions underlying the *GEBV* equations are required to be met. These assumptions are numerous and relate to several factors, such as the extent of coverage of genetic variability at QTLs by markers, which depends on the number and placement of markers in the genome. In addition, the estimation of the quality of markers is critical, as it is influenced by allele frequencies and the degree of linkage disequilibrium with QTLs, and may vary between populations, especially between reference and selection populations. Another crucial factor is the absence of nonadditive effects. The model rrBLUP is based on the additive relationship matrix (A) formed by the estimation of IBS as a marker-associated trait. Ridge regression is the main core of rrBLUP and is used to analyze genotypic data when they suffer from multicollinearity. With increasing marker density on genetic maps, the concept of multicollinearity is limited to strong LD, which is unusual for all genotypes between mono- and polymorphic SNPs. Therefore, even with ridge regression, unbiased least squares and large variance are observed in the GP results. Thus, the accuracy of model is affected by the training set, number of markers and heritability. Consequently, predicted genetic values are far from the actual values. The model gBLUP is based on the matrix A mating formed by the estimation of IBD and its relatives between the complex trait of interest and associated loci. The unavoidable and desired presence of LD between causal and/or marker loci modifies the simple interpretation of the matrices. The variability of LD in the genome due to heterogeneity within loci may lead to bias in the calculation of SNP heritability based on this genomic matrix. Both rrBLUP and gBLUP models are linear systems with an REML approach to predicting GS. Thus, the question of how to estimate genetic parameters such as variance–covariance components to calculate the fixed and random effects of high-density SNPs remains unanswered, and only point estimation of likelihood for *GEBV*s is available. However, GS linear models based on the REML approach can be specified without having to worry too much about the assumptions. However, Bayesian inference is more flexible when it comes to assumptions, and it has a range of answers that may change with each run [45]. The main challenge with BGLR and LASSO is to ensure that the distribution of statistical estimators follows the genetic parameters of the population. Therefore, it is often observed that kernel methods perform better compared to linear models. Kernel inference provides a linear solution in the feature space while being nonlinear in the input space, and it is a combination of linear and nonlinear parameters by definition. RKHS has partial similarity to rrBLUP and gBLUP models and utilizes a kernel matrix that represents Euclidean distances between focal points in Hilbert space. This kernel matrix in RKHS optimizes a more general structure of covariance between individuals compared to the GRM used to measure similarities in genetic values related to individuals. This allows greater flexibility in capturing complex relationships between individuals and improves the accuracy of predictions in GS. SVM can be interpreted as part of the class of kernel approaches. It is a method that is traditionally applied to classification problems, but has also been used for regression (prediction or selection). In SVM, optimization of the hyper-parameters can be carried out via various methods, but the most commonly used and convenient method is the grid search. The grid search method evaluates all possible combinations of hyper-parameters, allowing an exhaustive search in the hyper-parameter space. The BOOST, BAGG and STACK models are based on the DL algorithm with an ensemble method used to jointly solve a complex problem. A number of algorithms, generally nonparametric, are combined to improve the prediction or classification ability of the assembled model [46]. Our study has revealed that the definition and optimization of the regularization parameter is crucial to demonstrate the performance and accuracy of the GS model, which has not been sufficiently addressed in previous GS studies. However, in the BOOST model, the regularization parameter is only used to control the bias of the model. By adjusting the regularization parameter, the model can be made less complex and less prone to over-fitting. In the BAGG model, the regularization parameter is used to control the variance of the model. By reducing the variance, the model can be made more stable and less sensitive to noise. In STACK, both bias and variance are considered by adjusting the regularization parameter to find a balance between model complexity and stability. Thus, our study confirmed that the results of bias–variance tradeoff and adaptive prediction error for the STACK model were intermediate compared with other models. This remarkable result for the STACK model is consistent with previous results [23,47]. In all ensemble models, especially in the STACK, the number of epochs and batches of hyper-parameters need to be specified along with the activation process. The number of epochs determines how often the weights of model are updated, and it is affected by LR on the training dataset. It is important to carefully tune these hyper-parameters to optimize the performance of the model and avoid over-fitting or under-fitting. Therefore, a smaller LR in the training data set and a batch size of 1000 produces maximum SNP heritability and *GEBV* means. Thus, the epoch number indicates the forward and backward runs of the whole training data through the model. However, epoch number can be computationally intensive for the memory of computation, so these are divided into small batch sizes.

## 4. Materials and Method

### 4.1. Phenotypic Data

In this study, a set of 221 bread wheat genotypes from the Breeding Innovations in Wheat for Resilient Cropping Systems (BRIWECS) project were grown at the agricultural research station, Campus Klein-Altendorf, University of Bonn, Germany, in three cropping seasons during 2018, 2019 and 2020, in a split-plot design. The *NUE* value of each genotype under low-N (LN) and high-N (HN) fertilizers was calculated using the following formula:(3)$$NUE=\frac{GY}{N_{s}}=(\frac{N_{t}}{N_{s}})(\frac{GY}{N_{t}})$$ where GY is grain yield (gr/m^2^), $N_{s}$ is the nutrient supplied and $N_{t}$ is the total above-ground plant nutrient at maturity [48].

### 4.2. Genotypic Data

In order to characterize the *NUE* vector among the bread wheat population, a platform of 150K affymetrix SNP Chip at TraitGenetics GmbH (SGS GmbH Gatersleben, Germany), was used. To minimize monomorphism in the Chip, the SNPs with MAF ≤ 0.05 were removed. After checking for SNPs that deviated from the Hardy–Weinberg equilibrium (HWE), only 22,489 polymorphic SNP markers, were remained and were used in GS models.

### 4.3. Construction of GRM

Basically, the GRM is used as a kinship matrix in genome-wide association studies (GWASs) and GS models. For the rrBLUP model, based on the method suggested by [49], the covariance between individuals $g_{i}$ and $g_{j}$ can be equal to the covariance of ${SNP}_{ij}$; therefore, the GRM was calculated using $G=\frac{\Sigma_{k=1}^{L}{\left(g_{ik}-p_{k}\right)}^{2}}{\Sigma_{k=1}^{L}p_{k}\left(1-p_{k}\right)}$ and $p_{k}=\frac{1}{n}\Sigma_{i=1}^{n}g_{ik}$, where L is the number of loci, $p_{k}$ is the MAF for the loci k and $g_{i}$. In the rrBLUP model,$\;y=\mu +\sum_{i=1}^{p}X_{i}g_{i}+e$, where y is the *NUE* vector with n genotypes, p is the total number of SNPs, $X_{i}$ is the matrix of random SNP effects coded as (−1, 0 and 1), $g_{j}$ is the main diameter of GRM and $H_{0}$ denoted in the form of $\sigma_{g}^{2}=0$ based on identity by state (IBS) only, and e is the residuals of the model. In the gBLUP model, y=μ+Zu+e, where y is the *NUE* vector, Z is the matrix of genetic values for an individual, $V\left(u\right)=K\sigma_{g}^{2}$, where K is the GRM as a kinship matrix and $\sigma_{g}^{2}$ is the additive genetic variance with IBS or identity by descent (IBD). In the LASSO model, marker effects were calculated as $\hat{\beta }={\left(X^{*}X-\lambda I\right)}^{-1}X^{*}y$, where $\left[\lambda \right]=\frac{\sigma_{e}^{2}}{\sigma_{\beta }^{2}}$ is taken as a kinship matrix and I is an identity matrix. $\sigma_{\beta }^{2}$ and $\sigma_{e}^{2}$ are computed from SNP heritability and phenotypic variance, respectively [50,51]. The BGLR model with a Bayes factor was fitted to $y=\mu +X\alpha +Z\beta +{XZ}_{\alpha g}u+e$, where y is the *NUE* vector, X is a matrix with genotypes and SNPs, α is the corresponding vector of SNP effects which captures small effects of all SNPs, g is a vector that captures prior distribution of SNP effects with prior distribution of $g\sim N(0,\;G_{gu}^{2})$, where *G* is a marker-derived genomic relationship matrix, u is a vector of Bayes factors for the SNP matrix and e is the residual term [52,53]. In the RKHS, SVM, Boosting, Bagging and Stacking GS models, the objective is to classify SNPs with higher accuracy. Typically, in these models, the GRM with a linear scale has been replaced by a distance matrix with a Euclidean family scale, so the regularization parameter in any given GS model is a prerequisite for obtaining the kinship matrix.

### 4.4. Genomic Selection Models

1—In order to carry out GS based on frequentist and Bayesian perspectives, models including ridge regression best linear unbiased prediction (rrBLUP) with R/*rrBLUP* [54], genomic best linear unbiased prediction (gBLUP) with R/*BGLR* [55,56], least absolute shrinkage and selection operator (LASSO) with R/*glmnet* [45] and Bayesian generalized linear regression (BGLR) [48] with R/*BGLR* were applied. 2—To perform GS, models based on Kernel whole genome regression reproducing kernel Hilbert spaces (RKHS) with R/*BWGS* [57,58,59] and support vector machine (SVM) with R/*kernlab* [8,60] were utilized. 3—From tree ensemble algorithms, Boosting with R/*gbm* [61,62] and Bagging [33,63] with R/*ipred* were used. To run the Stacking GS model, [63,64] Python/*Scikit-learn* was utilized. All three category of GS models were run among the *NUE* vector of 221 wheat genotypes under low and high N levels.

### 4.5. Genetic Parameters and Hyper-Parameters Estimation

To solve the fixed effects coming from environmental factors or population structure, and random effects coming from SNPs, the residual maximum likelihood (REML) in all GS models were minimized through the BLUP vector as additive genomic variance and the BLUE vector as residual variances, and then SNP effects were estimated. The $h^{2}$, based on SNP information, was estimated as $h^{2}=\frac{\sigma_{a}^{2}}{{\sigma_{a}^{2}+\sigma }_{e}^{2}}$, where $\sigma_{a}^{2}$ and $\sigma_{e}^{2}$ are additive genomic and residual variances, respectively. In the CV strategy with the training set (K-fold = 10) and test set (K-fold = 5), genetic parameters, including *GEBV*s mean, bootstrap of *GEBV*s mean, genetic variance and error variance were applied to assess accuracy in the given GS model. In the CV approach, accuracy is measured using the correlation coefficient $r(\overset{ˇ}{y}Obs.,\overset{ˇ}{y}GEBV)$ between observed values (OBVs) and genomic estimated breeding values (*GEBV*s), which were estimated to divide the population into validation and training sets. The genotypes assigned to the validation set were used as predicted breeding values, and the remaining genotypes were used for the training set. Therefore, besides the genetic parameters in the definition of the GS models, before starting the training and testing processes, hyper-parameters were determined. They included learning rate (LR), number of hidden layers, number of iterations and batch size per one epoch computed. Based on the regularization parameters of a given model, the minimum learning rate was optimized with the rule $\alpha_{j}=\frac{100\alpha_{0}}{100+j}$, where $\alpha_{0}$ is the initial learning rate 1 and j is a counter of epochs to 9900 [65,66]. They were also defined based on SNP effects (*p*-values) generated from the *NUE* vector under low and high N content.

### 4.6. Bias–Variance Tradeoff in GS Models

Model evaluation was carried out for all GS models via analysis of bias and variance. The *NUE* vector was considered as a target trait at low and high N contents among all 221 bread wheat genotypes. Bias was taken as the difference between *GEBV*s and the true *NUE* vector. Variance was measured to identify the difference between parameters of a given model and its training set. In order to deal with model over- and under-fitting, the mean squared error (MSE) based on k-nearest neighbor (KNN) algorithm was calculated.

To reduce model complexity after dealing with outlier values in the *NUE* vector, irreducible error (IE) was measured.

### 4.7. Error Measurement between GS Models

To evaluate the GS models, additional error measurements were performed. Models based on *GEBV*s generated from the *NUE* vector under low and high N content into two pairwise groups were compared. To check the distribution of SNPs *p*-value, a matrix of standard errors (SE) was generated from each GS model under low and high N levels, after randomly sampling SNP effects 2000 times with replacement, under the null hypothesis *N* (0, 1), which is necessary when SNPs’ *p*-values are near to the normal distribution. For SNPs with *p*-values far from the normal distribution, the adaptive standard error of the prediction matrix was calculated with an adjusted FDR threshold of 0.05.

### 4.8. Genetic Selection Gain Estimation Based on the Selected Model

The equation $R=\frac{ir(\overset{ˇ}{y}Obs.,\overset{ˇ}{y}GEBV)}{y}\;log\left[\begin{array}{cc} X^{\prime }X & X^{\prime }Z\\ Z^{\prime }X & 1+\frac{1^{G^{*-1}}}{\sigma_{e}^{2}} \end{array}\right]$ was utilized to estimate the expected genetic selection gain (R). i is the selection intensity, $r(\overset{ˇ}{y}Obs.,\overset{ˇ}{y}GEBV)$ is the selection accuracy, y is the number of years. The component $log\left[\begin{array}{cc} X^{\prime }X & X^{\prime }Z\\ Z^{\prime }X & 1+\frac{1^{G^{*-1}}}{\sigma_{e}^{2}} \end{array}\right]$ is equal to $\beta_{i}$ which is the power of the given GS model. In this component, X is the incidence matrix for the proportion of individuals in the population structure (*n_ps_*) × marker (m) with fixed effect, $X^{\prime }$ is the transformed X, Z is a designed matrix for the effect of genotype (*n*) × marker effect (p), including all random effects, $Z^{\prime }$ is the transformed Z, $G^{*-1}$ is an invert matrix of the genomic relationship matrix (GRM), when the effect of non-associated markers are shrunken toward null with *N*(0,$\;\sigma_{e}^{2}$), and $\sigma_{e}^{2}$ is the covariate error of the GS model in the form of BLUEs. To clarify genetic gain from the GS model against gain from phenotypic selection, a relative efficiency (RE) index was developed. The RE of indirect GS under low and high N levels was calculated with ${RE}_{per\;N\;level}=\frac{r(\overset{ˇ}{y}Obs.,\overset{ˇ}{y}GEBV)}{\sqrt{R^{2}}}$.

## Figures and Tables

**Figure 1 ijms-24-14275-f001:**
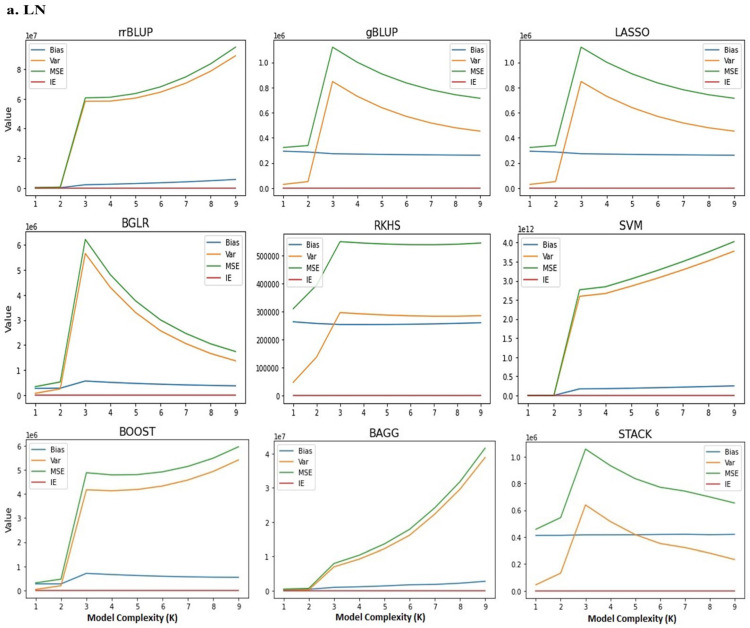
(**a**) Model complexity analysis using k-nearest neighbor (KNN) algorithm for the *NUE* vector under low N levels using different genomic selection models. X-axis represents value of bias (blue line), variance (orange line), MSE (green line) and irreducible error (red line); y-axis shows GS model complexity. le (number): ×10 ^number^. (**b**) Model complexity analysis using k-nearest neighbor (KNN) algorithm for the *NUE* vector under high N levels using different genomic selection models. X-axis shows the value of bias (blue line), variance (orange line), MSE (green line) and irreducible error (red line); y-axis shows GS model complexity, le (number): ×10 ^number^.

**Figure 2 ijms-24-14275-f002:**
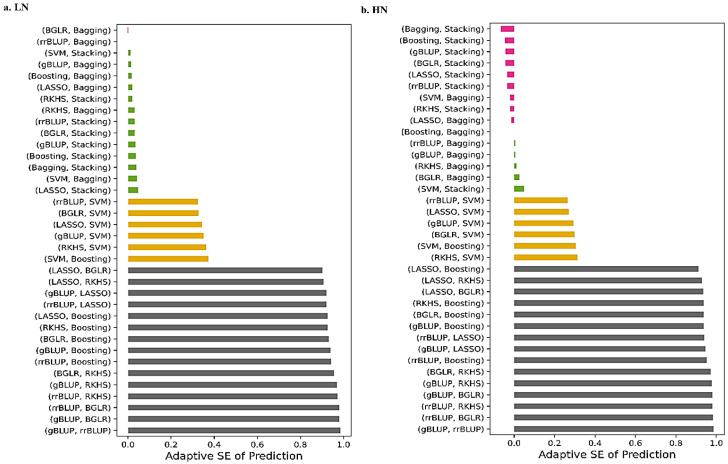
Adaptive standard error (SE) of prediction for pairwise comparison of GS models under low and high N levels. The ranges of adaptive SE values were normalized from −0.2 to 1. Pink color: the adaptive SE of prediction for the pairwise comparisons, is less than zero, green color: the adaptive SE of prediction for the pairwise comparisons, is equal by zero, orange color: the adaptive SE of prediction for the pairwise comparisons, is almost equal by 0.3 and gray color: the adaptive SE of prediction for the pairwise comparisons, is equal by 1.

**Figure 3 ijms-24-14275-f003:**
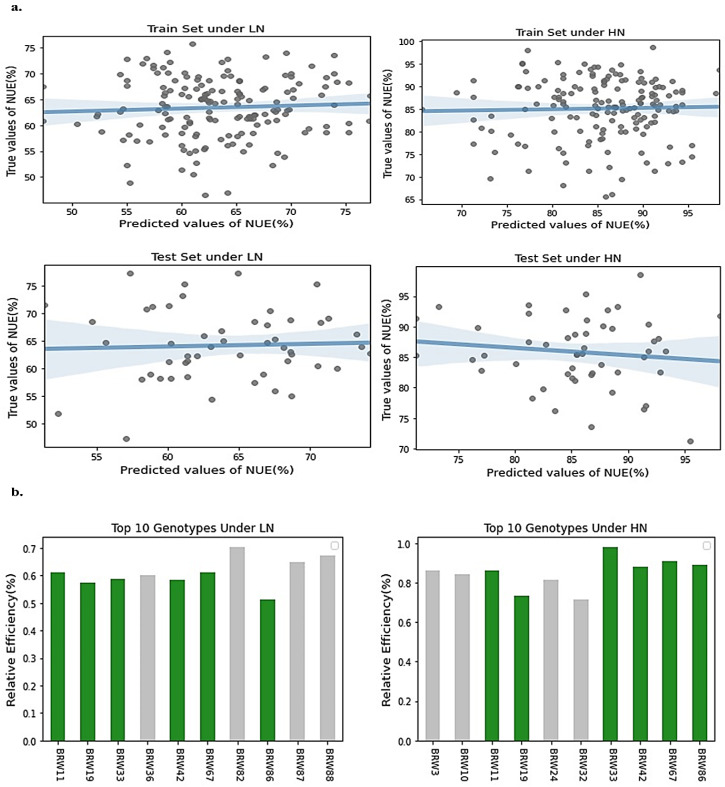
(**a**) Bias–variance analysis and adaptive SE of prediction indicate that the STACK model shows the best performance and accuracy. The predicted values of *NUE* (%) for both training and test sets under low and high N levels are displayed on the x-axis, while the true values from the dataset are displayed on the y-axis. (**b**) GS gain in the form of relative efficiency (%) for the STACK model among the 221 wheat genotypes and the top 10 genotypes under low and high N levels has been specified. Genotypes with green color under both low and high N levels are duplicate. Genotypes with gray color are distinct at each N level.

**Figure 4 ijms-24-14275-f004:**
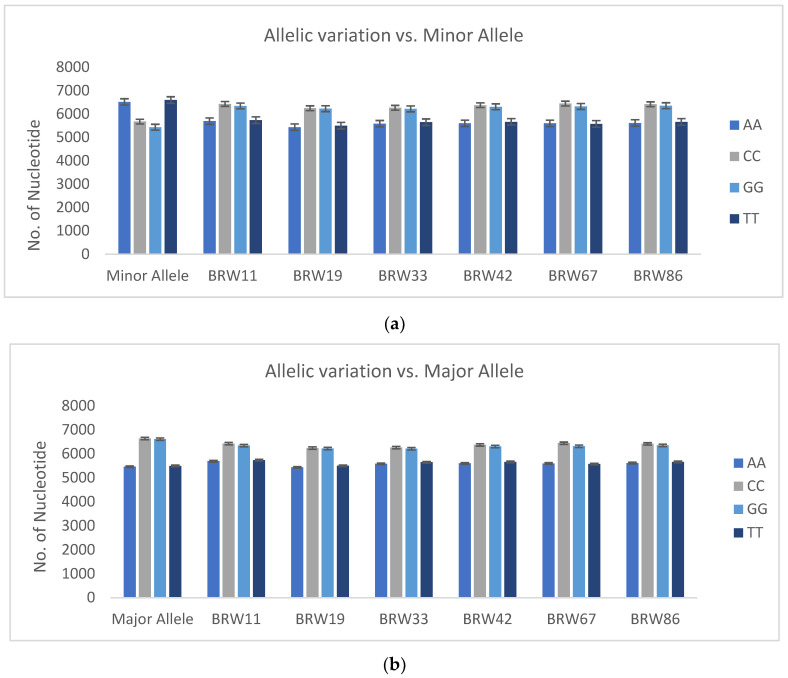
(**a**) Mean comparison between allele content of six top genotypes with highest GS gain for *NUE* (%) vector versus minor allele content in whole population. (**b**) Mean comparison between allele content of six top genotypes with highest GS gain for *NUE* (%) vector versus major allele content in whole population.

**Table 1 ijms-24-14275-t001:** Genetic parameter estimation for the *NUE* vector under low and high N levels using different genomic selection models.

Inference	Model	N Level	$\mathrm{SNP}-h^{2}$ ^a^	Training Set (CV_K_fold_ = 10)				Test Set (CV_K_fold_ = 5)				*p*-Value ^f^
				*GEBV*s Mean ^b^	Boot. *GEBV*s Mean ^c^	Vg ^d^	Ve ^e^	*GEBV*s Mean	Boot. *GEBV*s Mean	Vg	Ve	
Frequentist	rrBLUP	LN	0.30	0.65	0.66	27.49	64.15	0.64	0.66	28.77	67.13	0.05 *
		HN	0.33	0.73	0.70	9.79	19.88	0.73	0.70	8.77	17.81	
	gBLUP	LN	0.28	0.64	0.61	26.78	68.87	0.64	0.61	22.61	54.14	0.044 *
		HN	0.30	0.71	0.69	4.93	11.51	0.71	0.68	4.57	10.61	
Bayesian	LASSO	LN	0.31	0.62	0.61	32.40	72.12	0.62	0.60	32.03	71.12	0.144 ns
		HN	0.31	0.70	0.68	7.94	17.69	0.69	0.65	8.39	18.69	
	BGLR	LN	0.32	0.63	0.68	35.49	75.43	0.65	0.65	35.35	75.12	0.042 *
		HN	0.30	0.72	0.69	9.70	22.65	0.74	0.65	26.24	61.24	
Kernel	RKHS	LN	0.45	0.64	0.64	57.11	69.81	0.64	0.64	53.29	65.14	0.031 *
		HN	0.61	0.72	0.72	28.16	18.01	0.72	0.71	42.71	27.21	
	SVM	LN	0.38	0.13	0.22	24.30	39.66	0.18	0.32	24.64	40.21	0.048 *
		HN	0.57	0.18	0.29	73.09	55.14	0.18	0.33	59.66	45.01	
Ensemble	BOOST	LN	0.48	0.61	0.61	62.84	67.08	0.61	0.65	61.92	67.08	0.164 ns
		HN	0.62	0.68	0.69	28.73	17.61	0.68	0.72	27.76	17.02	
	BAGG	LN	0.55	0.60	0.68	71.03	58.12	0.61	0.71	69.82	57.13	0.679 ns
		HN	0.55	0.64	0.71	39.80	32.57	0.64	0.74	38.12	31.19	
	STACK	LN	0.62	0.69	0.78	49.33	30.24	0.69	0.79	50.85	31.17	0.0924 ns
		HN	0.71	0.76	0.78	72.98	29.81	0.76	0.79	73.49	30.02	

a—SNP-$h^{2}$: SNP-Heritability, b—*GEBV*s mean: mean of genomic estimated breeding values, c—Boot. *GEBV*s mean: Bootstrap mean of genomic estimated breeding values, d—V_g_: Genetic variance, e—V_e_: Error variance, f—*p*-value: H0: there is no significant difference between the means of genomic estimated breeding values (*GEBV*s) in training and testing sets under low and high N levels (α=0.05), significance levels of *p*-value * *p* ≤ 0.01, ns = not significant.

**Table 2 ijms-24-14275-t002:** Genetic hyper-parameter estimations for *NUE* vector under low and high N levels using different genomic selection models.

Inference	Model	N Level	Training Set (CV_K_fold_ = 10)				Test Set (CV_K_fold_ = 5)				*p*-Value ^b^
			LR ^a^	No. of Iteration	No. of Batch Size	Accuracy (%)	LR	No. of Iteration	No. of Batch Size	Accuracy (%)	
Frequentist	rrBLUP	LN	0.01	27	10	81.12	0.01	27	5	92.22	0.007 **
		HN	0.01	27	100	81.09	0.01	27	25	92.21	
	gBLUP	LN	0.001	27	10	80.88	0.001	27	5	92.21	0.001 **
		HN	0.001	27	100	80.41	0.001	27	25	92.22	
Bayesian	LASSO	LN	0.01	468	10	85.51	0.01	468	5	92.43	0.0024 *
		HN	0.01	468	100	85.12	0.01	468	25	92.44	
	BGLR	LN	0.001	468	10	84.47	0.001	468	5	92.74	0.0031 *
		HN	0.001	468	100	85.77	0.001	468	25	92.76	
Kernel	RKHS	LN	0.01	2050	100	85.11	0.01	2050	25	95.64	0.001 **
		HN	0.01	2050	1000	85.33	0.01	2050	250	95.37	
	SVM	LN	0.001	2050	100	84.04	0.001	2050	25	95.33	0.0014 **
		HN	0.001	2050	1000	85.77	0.001	2050	250	95.18	
Ensemble	BOOST	LN	0.01	5520	100	91.12	0.01	5520	25	96.01	0.098 ns
		HN	0.01	5520	1000	92.11	0.01	5520	250	96.12	
	BAGG	LN	0.001	5520	100	92.31	0.001	5520	25	96.48	0.1445 ns
		HN	0.001	5520	1000	92.16	0.001	5520	250	97.22	
	STACK	LN	0.001	5520	100	93.58	0.001	5520	25	97.54	0.0905 ns
		HN	0.001	5520	1000	93.79	0.001	5520	250	97.84	

a—LR: Learning rate of given GS model, b—*p*-value: H0: there is no significant difference between model accuracy in train and test sets under low and high N levels (α=0.05), significance levels of *p-*value * *p* ≤ 0.01, ** *p* ≤ 0.001, ns = not significant.

**Table 3 ijms-24-14275-t003:** Bias–variance tradeoff analysis for *NUE* vector under low and high N levels using different genomic selection models.

Inference	Model	Low N				High N			
		$\bar{Bias}$	$\bar{Var}$	$\bar{Exp.Loss}$	$\bar{Sk.Loss}$	$\bar{Bias}$	$\bar{Var}$	$\bar{Exp.Loss}$	$\bar{Sk.Loss}$
Frequentist	rrBLUP	9.1 × 10^5^	6.1 × 10^7^	6.2 × 10^7^	0.0009	0.2 × 10^6^	0.9 × 10^6^	1.1 × 10^6^	0.0009
	gBLUP	3.1 × 10^6^	0.8 × 10^6^	1.2 × 10^6^	0.0009	0.22 × 10^6^	3.7 × 10^6^	4.0 × 10^6^	0.0009
Bayesian	LASSO	3.1 × 10^6^	0.82 × 10^6^	1.2 × 10^6^	0.0009	0.2 × 10^6^	0.8 × 10^6^	1.2 × 10^6^	0.0009
	BGLR	3.1 × 10^6^	5.57 × 10^6^	6.4 × 10^6^	0.0009	0.2 × 10^6^	5.2 × 10^6^	5.5 × 10^6^	0.0009
Kernel	RKHS	25 × 10^4^	49 × 10^4^	55 × 10^4^	0.00081	0.3 × 10^6^	1.1 × 10^6^	1.5 × 10^6^	0.00081
	SVM	0.1 × 10^12^	2.5 ×1 0^12^	2.7 × 10^12^	0.00080	0.15 × 10^10^	1.5 × 10^10^	1.6 × 10^10^	0.00080
Ensemble	BOOST	4.1 × 10^6^	4.2 × 10^6^	5.1 × 10^6^	0.00007	0.09 × 10^7^	1.1 × 10^7^	1.19 × 10^7^	0.00007
	BAGG	0.75 × 10^7^	0.5 × 10^7^	0.8 × 10^7^	0.00007	0.11 × 10^7^	1.6 × 10^7^	1.75 × 10^7^	0.00007
	STACK	0.41 × 10^6^	0.6 × 10^6^	1.2 × 10^6^	0.00002	0.5 × 10^6^	5.1 × 10^6^	5.8 × 10^6^	0.00002
Mean		12.3 × 10^9^	30.0 × 10^9^	33 × 10^9^	0.00059	1.67 × 10^8^	1.68 × 10^9^	1.78 × 10^9^	0.00059
LSD (0.05)		0.92 × 10^9^	0.48 × 10^9^	1.25 × 10^9^	2.11	0.34 × 10^8^	0.70 × 10^9^	1.03 × 10^9^	2.11

$\bar{Bias}$: average of bias, $\bar{Var}$: average of variance, $\bar{Exp.Loss}$: average expected loss that is equal to mean square error (MSE) of GS model by bias–variance analysis simultaneously using k-nearest neighbor algorithm and $\bar{Sk.Loss}$: mean irreducible error (IE) of GS model by bias–variance analysis simultaneously using *Scikit-learn* algorithm. LSD (0.05) for comparison of GS models performances.

## Data Availability

The data that support the findings of this study are available from the corresponding author upon reasonable request.

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
