# Peer review of "Genetic Parameter and Hyper-Parameter Estimation Underlie Nitrogen Use Efficiency in Bread Wheat"

_ijms, 2023, doi:10.3390/ijms241814275_

Round 1
Reviewer 1 Report
In this study, 9 genomic selection methods were used. BSLMM (Bayesian Sparse Linear Mixed Models) is also a good prediction method with high accuracy. Do you think if it is better to add it in this study?
The authors mentioned cross-validation in genomic selection. Do you think if the replicates are needed to make the results more reliable?
The description of package for each method should be added in this paper.
Minor editing of English language required.
Author Response
We are very grateful for the reviews provided by the editors and each of the reviewers of this manuscript. It was your valuable and insightful comments that led to possible improvements in the current version. We have carefully considered the comments and tried our best to address every one of them. We have subjected our manuscript to extensive editing of English language and it has been completely rewritten again. We hope the manuscript after careful revisions meet your high standards. The authors welcome further constructive comments if any. Please, below we provide the point-by-point responses to the comments in blue.

Reviewer 2 Report
Figure 4 a and 4 b are unreadable. The legend has been cut out almost all of the figures.
There is no figure number for the figure comparing predictive values and true values of NUE. Also, there seems to be no relationship between the predicted NUE and the true NUE, so I have doubts about the usefulness of the model.
Almost every sentence needs to be rewritten. I would suggest that the authors use the spelling and grammar tools.
Author Response

(The authors gave the same response as above.)
